

**Ammonia impacts methane oxidation and methanotrophic community in**

**freshwater sediment**

Yuyin Yang[1], Jianfei Chen[1], Shuguang Xie[1,*], Yong Liu[2]

[1]State Key Joint Laboratory of Environmental Simulation and Pollution Control,

College of Environmental Sciences and Engineering, Peking University, Beijing

100871, China

[2]Key Laboratory of Water and Sediment Sciences (Ministry of Education), College of

Environmental Sciences and Engineering, Peking University, Beijing 100871, China

* Corresponding author. Tel: 86-10-62751923. Email: xiesg@pku.edu.cn (S Xie)



**Abstract**
Lacustrine ecosystems are an important natural source of greenhouse gas methane.
Aerobic methanotrophs are regarded as a major regulator controlling methane
emission. Excess nutrient input can greatly influence carbon cycle in lacustrine
ecosystems. Ammonium is believed to be a major influential factor, due to its
competition with methane as the substrate for aerobic methanotrophs. To date, the
impact of ammonia on aerobic methanotrophs remains unclear. In the present study,
microcosms with freshwater lake sediment were constructed to investigate the
influence of ammonia concentration on aerobic methanotrophs. Ammonia influence
on the abundance of *pmoA* gene was only observed at a very high ammonia
concentration, while the number of *pmoA* transcripts was increased by the addition of
ammonium. *pmoA* gene and transcripts differed greatly in their abundance, diversity
and community compositions. *pmoA* transcripts were more sensitive to ammonium
amendment than *pmoA* gene. Methane oxidation potential and methanotrophic
community could be impacted by ammonium amendment. This work could add some
new sights towards the links between ammonia and methane oxidation in freshwater
sediment.

**Keywords:** Ammonium; Freshwater lake; Methane oxidation; Methanotroph; *pmoA*
gene; *pmoA* transcripts




## 1. Introduction


Methane is a major product of carbon metabolism in freshwater lakes, and also a


critical greenhouse gas in the atmosphere (Bastviken et al., 2004). Aerobic methane


oxidation performed by bacterial methanotrophs is a major pathway controlling


methane emission. Up to 30–99% of the total methane produced in anoxic sediment


can be oxidized by methanotrophs (Bastviken et al., 2008). Methane oxidation in


freshwater lakes can be greatly influenced by the environmental changes (e.g.


eutrophication) induced by anthropogenic activities (Borrel et al., 2011).



The increasing input of nutrients into freshwater lakes has greatly raised the


availability of dissolved organic carbon (DOC), nitrogen and phosphorus, and also


exerted a considerable influence on aerobic methane oxidation (Liikanen and


Martikainen, 2003; Veraart et al., 2015). Among various types of nutrients, ammonia


has attracted great attention. Ammonium and methane have similar chemical


structure, and ammonium is known to compete with methane for the binding site of


methane monooxygenase, a key enzyme in methane oxidation (Bédard and Knowles,


1989). On the other hand, a high concentration of oxygen in lake water might also


inhibit methane oxidation (Rudd and Hamilton, 1975), and excess ammonia can lead


to the competition between methane oxidizers and ammonium oxidizers for oxygen.


With high oxygen availability or low in-situ nitrogen content, methane oxidation can




also be stimulated by the addition of ammonium (Rudd et al., 1976). Hence, the effect
of ammonium on methane oxidation is complex (Bodelier and Laanbroek, 2004), and
previous studies have documented contradictory results, such as inhibition (Bosse et
al., 1993; Murase and Sugimoto, 2005; Nold et al., 1999), no effect (Liikanen and
Martikainen, 2003), or stimulation (Bodelier et al., 2000; Rudd et al., 1976). The
effect of ammonium on methane oxidation might largely depend on the characteristics
of the studied ecosystem and in-situ environment (Bodelier and Laanbroek, 2004;
Borrel et al., 2011).

To date, previous studies about the ammonium effect on methane oxidation in
freshwater lakes mainly focused on either oxidation rate or net methane flux (Bosse et
al., 1993; Liikanen and Martikainen, 2003; Murase and Sugimoto, 2005). However,
methanotrophs play a fundamental role in regulating methane emission from
freshwater sediment (Bastviken et al., 2008). The abundance, transcription, and
community structure of aerobic methanotrophs may also be affected by the extra input
of ammonium (Shrestha et al., 2010). The difference in methanotrophic community
structure can further lead to various responses of methane oxidation to nitrogen
content (Jang et al., 2011; Mohanty et al., 2006; Nyerges and Stein, 2009). Therefore,
identification of the variation of methanotrophic community can be helpful to
understand how ammonium input influences methane oxidation process. The
community change of methanotrophs under ammonium stress has been observed in





various soils, such as agriculture soil (Seghers et al., 2003; Shrestha et al., 2010) and
landfill soil (Zhang et al., 2014). The results of these previous studies suggested that
the effect of ammonium on methanotroph community might be habitat-related. A
recent filed work suggested that in-situ ammonia concentration might be a key
regulating factor of methanotrophic community structure in freshwater lake sediment
(Yang et al., 2016). However, the direct evidence for the influence of ammonium (or
ammonia) on methanotroph community in freshwater lake sediment is still lacking.
Hence, in the present study, microcosms with freshwater lake sediment were
constructed to investigate the ammonium influence on methane oxidation potential
and the abundance, transcription and community structure of aerobic methanotrophs.

**2. Materials and methods**
*2.1. Sediment characteristics*
Dianchi Lake is a large shallow lake (total surface area: 309 km$^2$; average water
depth: 4.4 m) located in southeast China (Yang et al., 2016). This freshwater lake is
suffering from anthropogenically-accelerated eutrophication (Huang et al., 2017).
Surface sediment (0–5 cm) (24.9286N, 102.6582E) were collected using a core
sampler from the north part of Dianchi Lake in October, 2017. In-situ dissolved
oxygen (DO) and ammonium nitrogen (NH$_4^+$-N) in overlying water were 8.37 mg/L
and 344 μM, respectively. Sediment total organic carbon (TOC), total nitrogen (TN),
the ratio of TOC to TN (C/N), nitrate nitrogen (NO$_3^-$-N), ammonium nitrogen (NH$_4^+$-



N), total phosphorus (TP), and pH were 41.3 g/kg, 3.95 g/kg, 10.5, 12.3 mg/kg, 364
mg/kg, 0.60 g/kg, and 7.2, respectively. Sediment (2 L) was transported to laboratory
at 4ºC for incubation experiment.

*2.2. Experimental setup*
Sediments were placed at room temperature for 24 h and then homogenized. The
homogenized sediments were centrifuged at 5000 rpm for 10 min to determine the
initial ammonia concentration of pore water. A series of 50-mL serum bottles (as
microcosms) were added with 10 mL of sediment aliquot (containing about 0.1 g dry
sediment). A total of 111 microcosms were constructed, including three autoclaved
ones used as the control for the measurement of methane oxidation potential. Six
treatments (A–F) were set up. The microcosms with treatments B–F were added with
1 mL of $NH_4Cl$ at the levels of 5, 20, 50, 100, and 200 mM, respectively, while the
microcosm with treatment A was amended with 1 mL diluted water as the blank
control. For each treatment, 18 microcosms were constructed, including half used for
molecular analyses and another half for methanotrophic potential measurement. These
microcosms were closed with butyl rubber stoppers and incubated for 14 days at 25ºC
at 100 rpm in dark.

At each sampling time point (day 1 (12 h after incubation), day 7 or day 14), triplicate
sediment samples of each treatment were transferred into Falcon tubes, and then




centrifuged at 5000 rpm for 10 min. The supernatant was filtered with a 0.2-μm
syringe filter, and its ammonia level was measured using Nessler reagent-colorimetry.
The sediment was mixed up and immediately used for nucleic acid extraction. In
addition, at each sampling time, for methanotrophic potential measurement, another
three bottles of each treatment were opened and shaken to provide ambient air, then
closed again with butyl rubber stoppers. Headspace air (1 mL) was replaced by $CH_4$
(99.99%) with an air-tight syringe. Samples were shaken vigorously to mix. After
incubation at 25ºC, 100 rpm for 24 h, 0.1 mL of headspace gas was taken and
measured using a GC126 gas chromatograph equipped with a flame ionization
detector. Autoclaved control was also processed to exclude methane loss due to
dissolution or airtightness.

*2.3. Nucleic acid extraction, reverse transcription and quantification*
Sediment DNA and RNA were extracted with PowerSoil DNA Isolation Kit (MoBio)
and PowerSoil Total RNA Isolation Kit (MoBio, USA), respectively. The quality and
concentration of extracted nucleic acids were examined with Nanodrop 2000 (Thermo
Fisher Scientific, USA). RNA was diluted to a similar concentration before further
analysis. Real-time PCR of *pmoA* gene was performed on a CFX Connect cycler
(Bio-Rad, USA), using the primer set A189f/mb661r following the conditions
reported in our previous study (Liu et al., 2015). Reactions were carried out using a
TransStart Top Green qPCR Kit (Transgen, China) following the manufacturer's



instructions. Gene transcripts were quantified in a one-step RT-qPCR using a
TransScript Green One-step qRT-PCR Kit. Melting curve analyses were carried out at
the end of PCR run to check the amplification specificity. Each measurement was
carried out with three technical replicates. Standard curve was constructed with *pmoA*
gene clones, and the efficiency and r-square were 91.5% and 0.998, respectively.

*2.4. Terminal restriction fragment length polymorphism (T-RFLP) fingerprinting*
DNA *pmoA* gene fragment was amplified with primer sets A189f/mb661r, with the
forward primer A189f modified with FAM at 5'-end. PCR reactions were performed
as previously described (Liu et al., 2015). Two-step RT-PCR was carried out on
RNA. In the first step, RNA was reversely transcribed into cDNA with *pmoA* gene
specific primer using One-step gDNA removal and cDNA synthesis kit (Transgen
Biotech Co., LTD, China). The 20-μL reaction solution contained 1 μL EasyScript
RT/RI Enzyme Mix, 1 μL gDNA remover, 10 μL 2×ES Reaction Mix, 2 pmol of gene
specific primers and 1 μL RNA template. The reaction mixture was incubated at 42ºC
for 30 min, and the enzymes were deactivated at 85ºC for 5 s. In the second step, 1 μL
cDNA was used as template in *pmoA* gene PCR amplification, proceeded following
the same protocol with DNA.

The fluorescently labeled PCR products were purified using a TIANquick Mini
Purification Kit (TIANGEN Bitotech Co., Ltd, China). Approximately 20 ng of





purified PCR products were digested with restriction endonuclease *BciT130* I (Takara
Bio Inc., Japan) following the conditions recommended by the manufacturer's
instruction. Electrophoresis of digested amplicons was carried out by Sangon Biotech
(China) using an ABI 3730 DNA analyzer (Thermo Fisher Scientific, USA). The
length of T-RFs was determined by comparing with internal standard using the
GeneScan software. Terminal restriction fragments (T-RFs) with similar length (less
than 2 bp difference) were merged, and T-RFs shorter than 50 base pairs (bp) or
longer than 508 bp were removed from the dataset. Relative abundance of each
fragment equaled to the ratio of its peak area to the total area. Minor T-RFs with
relative abundance less than 0.5 % were excluded for further analysis. The Shannon
diversity indices of *pmoA* gene and transcripts were calculated based on DNA and
RNA T-RFs, respectively.

*2.5. Cloning, sequencing and phylogenetic analysis*
*pmoA* gene clone library was generated with mixed DNA PCR products using a TA
cloning kit (TransGen Biotech Co., LTD, China). Randomly picked clones were
subjected to sequencing. The in silico cut sites of these *pmoA* sequences were
predicted using the online software Restriction Mapper
(http://www.restrictionmapper.org). The sequences of each T-RF, together with their
reference sequences from the GenBank database, were used for phylogenetic analysis.
A neighbor-joining tree was conducted with MEGA 7 (Kumar et al., 2016), and





bootstrap with 1000 replicates was carried out to check the consistency. The
phylogenetic tree was visualized using iTOL v4.2 (Letunic and Bork, 2016). The
sequences used in phylogenetic analysis were deposited in GenBank database, and the
accessions were shown in Fig. 3.

*2.6. Statistical analysis*
Two-way ANOVA (analysis of variance) was carried out to determine the effect of
ammonia concentration and incubation time on $CH_4$ oxidation potential, gene
abundance and transcription. One-way ANOVA followed by Student-Newman-Keuls
test was adopted to detect the difference among treatments. The analysis was carried
out in *R*, using *R* packages stats (version 3.4.4) and agricolae (version 1.2-8).
Moreover, the comparison of methanotrophic communities in different microcosms,
using Redundancy Analysis (RDA) and clustering analysis, was carried out with *R*
package Vegan (version 2.4-6) (Oksanen et al., 2018). Permutation test was carried
out to detect the margin effect of variables (treatment and time). Clustering analysis
was carried out based on Bray-Curtis dissimilarity, to demonstrate the variation of
microbial community structure during incubation.

**3.  Results**
*3.1.  Methane oxidation potential*





210 Ammonium was found to quickly deplete in each ammonium added microcosm (Fig.

211 S1). Methane oxidation potential (MOP) varied from 0.77 (in the microcosm with

212 treatment F on day 1) to 1.94 (in the microcosm with treatment F on day 14) mmol/g

213 dry sediment day (Fig. 1), while autoclaved control did not show notable methane

214 oxidation (data not shown). Based on two-way ANOVA, both ammonium

215 concentration (treatment) and incubation time had significant effects on MOP ($P<$

216 0.01), and their interaction was also significant ($P< 0.05$). The MOP in the microcosm

217 with treatment A (with no external ammonium addition) did not show a significant

218 difference among incubation times ($P>0.05$). Based on post-hoc test (Fig. 1, Table

219 S1), at each time, the microcosm with treatment B had slightly higher MOP than the

220 microcosm with treatment A. At days 1 and 7, the microcosms with treatments C, D

221 and E had slightly lower MOP than the un-amended microcosm. However, at each

222 time, no statistical difference in MOP was observed among the microcosms with

223 treatments A–E. Moreover, the microcosm with treatment F (with the highest

224 ammonium addition) tended to have significantly lower MOP than other microcosms

225 on day 1 ($P< 0.05$), but significantly higher MOP on day 14 ($P< 0.05$). On day 7, no

226 statistical difference in MOP was found between the microcosm with treatment F and

227 any other microcosms.


229 *3.2. pmoA gene and transcript abundance*



Two-way ANOVA indicated that the number of both *pmoA* gene and transcripts was
significantly influenced by ammonium concentration and incubation time ($P < 0.01$)
(Fig. 2a and 2b). The abundance of *pmoA* gene in the microcosm with treatment A
showed no significant difference among times ($0.05 < P < 0.1$). On day 1, the
microcosms with treatments C and D had higher (but not significantly) *pmoA* gene
abundance than other microcosms. However, at days 7 and 14, the microcosm with
treatment F (with the highest ammonium addition) had the highest *pmoA* gene
abundance.

At each time, *pmoA* transcripts in the un-amended microcosm was less abundant than
those in amended microcosms. On day 1, the highest number of transcripts was
observed in the microcosm with treatment C, followed by the microcosms with
treatments D, E and F. The microcosm with treatment B had much lower *pmoA*
transcript abundance than other ammonium added microcosms ($P < 0.05$) (Table S1).
On day 7, *pmoA* transcript abundance tended to increase with the level of added
ammonium, although statistical difference in *pmoA* transcript abundance was only
observed between treatment F and other treatments. On day 14, no significant
difference in *pmoA* transcript abundance was detected among treatments ($P > 0.05$).

The ratio of transcripts to *pmoA* gene varied with ammonium concentration and
incubation time (Fig. S2). The ratio tended to decrease with time in ammonium





amended microcosms. Moreover, at days 1 and 7, the ratio tended to increase with the
increasing ammonium concentration.

*3.3. T-RFLP fingerprinting*
In silico analysis of the cloned *pmoA* sequences showed that restriction enzyme
*BciT130* I could well capture *pmoA* gene diversity and present a good resolution
among different subgroups of aerobic methanotrophs. Most of the T-RFs retrieved in
the current study could be assigned to certain methanotrophic groups, while some of
the T-RFs from *pmoA* transcripts could not match the cut site predicted from the
sequences in clone library. The obtained *pmoA* sequences could be grouped into four
clusters (Fig. 3), which could be convincingly affiliated with known methanotrophic
organisms. Three clusters were affiliated with Type I methanotrophs
(*Gammaproteobacteria*), which could be further divided into several subgroups.
Cluster 1 contained 157 bp, 242 bp and 338 bp T-RFs that could be related to Type Ia
methanotrophs, the most frequently detected methanotrophs in freshwater lakes
(Borrel et al., 2011). The 157 bp and 338 bp T-RFs might be affiliated with
*Methylobacter* and *Methylomicrobium*, respectively. However, the 242 bp T-RF could
not be convincingly assigned to a certain genus because of the highly similar *pmoA*
sequences of Type Ia organisms. Cluster 2 was composed of three different T-RFs,
and could be affiliated with *Methylococcus* and *Methyloparacoccus*. Cluster 3
included the T-RFs of 91 bp and 508 bp, which might be closely related to



*Candidatus Methylospira*. Both cluster 2 and cluster 3 could be affiliated with Type
Ib methanotrophs, but they distinctly differed in phylogeny and morphology
(Danilova et al., 2016). Cluster 4 comprised of the T-RFs of 217 bp, 370 bp and 403
bp, and it was phylogenetically related to Type IIa methanotrophs (*Methylocystaceae*
in *Alphaproteobacteria*). The 403 bp T-RF was likely affiliated with *Methylosinus*,
while 217 bp and 370 bp T-RFs could not be convincingly assigned to a single genus.

The 508 bp fragment could be affiliated with either *Methylospira* or unknown Type Ia
methanotroph. Considering the low abundance of 508 bp T-RF (<0.5% in DNA
TRFLP profile and approximately 2% in RNA TRFLP profile), and in order to avoid
incorrect annotation, this T-RF was excluded from further analysis.

*3.4. T-RFLP diversity and profiles of pmoA gene and transcripts*
Diversity of each community was calculated based on T-RFLP results. In the current
study, the T-RFs with relative abundance more than 5% in at least one sample or with
average relative abundance more than 2% in all samples were defined as major T-
RFs. For a given sample, the total number of T-RFs and the number of major T-RFs
were greater in RNA T-RFLP profile than in DNA T-RFLP profile. On day 1,
ammonium amended microcosms tended to have lower *pmoA* gene diversity than un-
amended microcosm, while an opposite trend was found at days 7 and 14 (Table 1).
For a given sample, *pmoA* transcript showed higher Shannon diversity than *pmoA*




293 gene. Ammonium amended microcosms tended to have lower *pmoA* transcript

294 diversity than un-amended microcosm. In the microcosms with treatments A–D,

295 *pmoA* transcript diversity tended to increase with time. However, the Shannon

296 diversity of transcriptional T-RFs experienced an increase followed by a decrease in

297 the microcosms with treatments E and F.

298

299 A total of 11–14 T-RFs were retrieved from T-RFLP analysis of DNA samples. Most

300 of them (including all major T-RFs) could be well assigned to certain methanotrophic

301 groups (Figs. 3 and 4a). In all DNA samples, Type Ia and Type IIa methanotrophs

302 dominated methanotrophic communities. On day 1, the 242 bp T-RF (*Methylobacter*-

303 related Type Ia methanotrophs) comprised about 50% of methanotrophic

304 communities. The 370 bp T-RF (Type IIa methanotrophs) also showed a considerable

305 proportion (20–25%). The addition of ammonium tended to induce no considerable

306 change of methanotrophic community structure after 12-hour incubation. After 7 and

307 14 days of incubation, the proportions of major T-RFs illustrated an evident variation.

308 The proportion of Type Ia methanotrophs (157 bp, 242 bp and 338 bp; marked in

309 green) decreased with time, while Type IIa methanotrophs (217 bp and 370 bp,

310 marked in pink) increased. The proportion of *Methylococcus*-related Type Ib

311 methanotrophs (marked in blue) also increased, especially the 145 bp T-RF, whereas

312 the proportion of *Methylospira*-related Type Ib methanotrophs (91 bp, marked in

313 yellow) did not show a notable variation.






A total of 14–38 T-RFs were retrieved from T-RFLP analysis of RNA samples, but
most of them were only detected in a few samples with low relative abundance (Fig.
4b). Among the major transcript T-RFs, only 4 transcript T-RFs could be assigned to
a known methanotrophic group, and on day 1 they comprised of a considerable part of
methanotrophic community in un-amended microcosm (43–72%) and of in amended
microcosms (22–72%), while the other 7 T-RFs were not found in *pmoA* gene library
as well as DNA T-RFLP profiles. Compared with *pmoA* gene, the community
structure of *pmoA* transcripts was more sensitive to external ammonium addition. The
addition of ammonium induced a marked shift in *pmoA* transcriptional community
structure after 12-h incubation. The proportion of 242 bp increased, but the proportion
of 91 bp decreased. After 1 or 2 weeks' incubation, the microcosms with treatments
B, C, D and E had similar transcriptional community structure as the un-amended
microcosm. However, the microcosm with treatment F (with the highest ammonium
addition) encountered a remarkable increase in 91 bp (Ca. *Methylospira*-related Type
Ib methanotrophs). Moreover, the 370 bp T-RF, accounting for up to one fourth
(average) of DNA T-RFs, was only detected on day 14, with relative abundance of
0.8–2.9%.

*3.5. Clustering and statistical analysis of TRFLP profiles*



DNA- and RNA-based methanotrophic community structures were characterized with
hierarchal clustering based on Bray-Curtis dissimilarity (Fig. 5). *pmoA* community
structure was quite stable during the whole incubation period. Most of the samples on
day 1 were grouped together. Samples B7, D7, E7, B14, C14, D14, E14 and F14 were
clustered into another group. Sample D1 was distantly separated from other samples,

Higher dissimilarity of transcriptional community structures could be observed among
samples. The samples on day 1 were still close to each other, and they were clearly
separated from the samples at day 7 and 14. Samples A7, A14, B7, B14, D7, E7 and
F7 could form a clade, while samples C7, D14 and E14 formed another clade.
Moreover, sample F14 was distantly separated from other samples.

RDA with permutation test was carried out to test the potential relationship between
each major T-RF and factors (treatment and incubation time). The result indicated that
incubation time had a significant impact on DNA-based methantrophic community
composition ($P< 0.01$), while ammonium concentration did not exert a significant
influence ($P>0.05$). The constrained variables could explain up to 74.4 % of total
variance. However, most of the explained variance (73.7% out of 74.4%) was related
to constrained axis 1, and only the first axis was significant ($P= 0.029$). In addition,
for RNA-based methantrophic community, treatment and time were able to explain
76.0% of total variance. Only incubation time had a significant effect on RNA-based





methantrophic community composition ($P< 0.01$), and only the first constrained axis
was significant ($P<0.01$). These results indicated that after the addition and with the
depletion of ammonium, the community compositions of both *pmoA* gene and
transcripts could undergo a considerable shift.

**4.  Discussion**
*4.1.  Effect of ammonium on MOP*
The current study showed that a high dosage of ammonium could present a temporary
inhibition effect on methane oxidation. The result was consistent with several
previous studies (Bosse et al., 1993; Murase and Sugimoto, 2005; Nold et al., 1999).
These studies indicated that the addition of ammonium might inhibit methane
oxidation in water and sediment of freshwater lake. However, to date, the minimal
inhibit concentration for methane oxidation in lake sediment is still unclear. Bosse et
al. (1993) pointed out that methane oxidation in littoral sediment of Lake Constance
could be partially inhibited when ammonium concentration in pore water was higher
than 4 mM. In contrast, methane oxidation in sediment of hyper-eutrophic Lake
Kevätön was not obviously affected by a continuous water flow containing up to 15
mM of ammonium (Liikanen and Martikainen, 2003). Lake Kevätön and Dianchi
Lake had similar average water depth, and the overlying water of sediment in both
lakes had very high levels of ammonium (Liikanen and Martikainen, 2003). In the
present study, inhibition was only observed in the microcosm with a very high



ammonium dosage (with 17.3 mM ammonium in overlying water on day 1), while no
evident inhibition was found in the other ammonium amended microcosms, even at
high dosages. This suggested that methane oxidation might depend on ammonium
dosage, which was sustained by the result of two-way ANOVA. The minimal inhibit
concentration for methane oxidation in Dianchi Lake was much higher than that in
Lake Constance (Bosse et al. 1993). Hence, the minimal inhibition concentration for
methane oxidation could be lake-specific.

Despite of a very high dosage of ammonium, sediment MOP was only partially
inhibited. This might be explained by two facts. The studied sediment sample
originated from a eutrophic lake, which suffered from high ammonium input.
Methanotrophs in this kind of ecosystem could effectively oxidize methane under the
condition of high ammonia concentration (Liikanen and Martikainen, 2003). This was
consistent with the above-mentioned lake-related minimal inhibition concentration for
methane oxidation. On the other hand, the affinity of pMMO (*pmoA* encoding
protein) to methane is much higher than that to ammonium (Bédard and Knowles,
1989). As a result, when the methane concentration is high enough, as a common case
for the measurement of MOP, methanotrophs should be able to consume a
considerable amount of methane.





A recovery of MOP after a single-shot fertilization has been reported in forest soil
(Borjesson and Nohrstedt, 2000). In this study, it was noted that after the depletion of
ammonium, sediment MOP could also get a quick recovery. The highest ammonium
dosage eventually stimulated sediment MOP in the long run (about two weeks).
Considering the increase of *pmoA* gene abundance and the change of RNA-based
methantrophic community structure, this might be attributed to an adaption to the
environment. The initial decrease of MOP could be explained by the competition
between methane and ammonium for pMMO (Bédard and Knowles, 1989), while the
subsequent increase of MOP might be the consequence of the shift in methantrophic
community structure (Seghers et al., 2003; Shrestha et al., 2010) and the increase of
*pmoA* gene abundance and transcription.

*4.2. Effect of ammonium on pmoA gene and transcript abundance*
So far, little is known about the changes of methanotrophic abundance and transcripts
induced by external ammonium amendment. Alam and Jia (2012) reported that the
addition of 200 μg of nitrogen/g dry weight soil (in ammonium sulfate) showed no
significant influence on *pmoA* gene abundance in paddy soil. However, in
ammonium-amended rhizospheric soil microcosms, *pmoA* gene abundance slightly
increased after 29 days' incubation (Shrestha et al., 2010). In this study, after 7 days'
incubation, the sediment microcosm with the highest ammonium dosage had much
higher *pmoA* gene abundance than un-amended microcosm and other amended

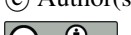



microcosms with lower dosage, whereas no significant difference of *pmoA* gene
abundance was detected between un-amended microcosm and amended microcosms
(except for the treatment with the highest ammonium dosage). After 14 days'
incubation, the microcosm with the highest ammonium dosage also had much higher
DNA-based methanotrophic abundance than other amended microcosms. Hence, the
present study further provided the evidence that the addition of ammonium,
depending on dosage, could influence freshwater sediment DNA-based
methanotrophic abundance, which was in agreement with the result of two-way
ANOVA. Dianchi Lake had been suffering from eutrophication for over 30 years
(Huang et al., 2017). It could be assumed that methanotroph community in this lake
had been adapted to high in-situ ammonia concentration. As a result, only extremely
high dosage of ammonium could pose a significant impact on DNA-based
methanotrophic abundance.

At each time, the microcosm with no external ammonium addition had lower
abundance of *pmoA* transcripts than each amended microcosm. This suggested the
addition of ammonium could influence the transcription of *pmoA* gene. The
stimulation of *pmoA* transcription by the addition of ammonium could be attributed to
the competition between methane and ammonium for the binding site of pMMO
(Bédard and Knowles, 1989). This was also verified by the similar number of
transcripts in these amended microcosms after the considerable reduction of





ammonium. At days 1 and 14, the abundance of *pmoA* transcripts differed greatly in
different amended microcosms. This suggested that ammonium dosage could
influence the number of *pmoA* transcripts, which was consistent with the result of
two-way ANOVA.

*4.3. Effect of ammonium on DNA- and RNA-based* methanotrophic *community*
*compositions*
Several previous studies have investigated the influence of ammonium amendment on
soil methanotrophic community structure (Alam and Jia, 2012; Mohanty et al., 2006;
Shrestha et al., 2010), yet information about the influence of ammonium amendment
on freshwater methanotrophic community structure is still lacking. In this study,
immediately after ammonium addition (after 12-h incubation), the relative abundance
of Type I (especially Type Ia) methanotrophs transcripts increased, instead of Type II.
This coincided with the result reported in rice and forest soils (Mohanty et al., 2006).
This also suggested that a high level of ammonia favored the growth of Type I
methantrophs and they might play an important role in methane oxidation in
ammonia-rich lake. However, both DNA- and RNA-based T-RFLP profiles indicated
that the addition of ammonium lead to an increase in the ratio of Type II to Type I
methanotrophs in two weeks, which was contrary to the results observed in some
previous studies in soil ecosystems (Alam and Jia, 2012; Bodelier et al., 2000;
Mohanty et al., 2006). These previous studies found that Type I methanortrophs had a





numerical advantage over Type II at a high ammonia concentration. Our recent field
study suggested that that the abundance of Type II methanotrophs in sediment of
Dianchi Lake was closely correlated to the concentration of ammonia (Yang et al.,
2016). Therefore, the response of methanotrophs to ammonia might depend on the
type of ecosystem. In addition, the community compositions of *pmoA* genes and
transcripts could be divergent, and DNA-based and RNA-based methanotrophs could
show different responses to ammonium addition (Shrestha et al. 2010). In this study,
compared with *pmoA* gene, the community structure of *pmoA* transcripts was more
sensitive to external ammonium addition. This was in a consensus with the result of a
previous study on the effect of ammonium addition on methanotrophs in root and
rhizospheric soils (Shrestha et al. 2010).

**5. Conclusions**
This was the first microcosm study on the influence of ammonium on freshwater lake
sediment methanotroph community. In freshwater lake sediment microcosm, methane
oxidation potential and methanotrophic community could be influenced by
ammonium amendment. Ammonia concentration had a significant impact on
methanotrophic abundance and diversity, but exerted no evident influence on
community structure. Compared with *pmoA* gene, transcripts were more sensitive to
external ammonium addition. Further works are necessary in order to elucidate the
influence of ammonium on methane oxidation in freshwater sediment.




**Conflict of interest**

The authors declare that they have no competing interests.


**Acknowledgments**

This work was financially supported by National Natural Science Foundation of

China (No. 41571444), and National Basic Research Program of China (No.

2015CB458900).

488

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






**Table 1** Numbers of T-RFs and T-RF-based Shannon diversity. For sample name,
upper case letters refer to treatment while digits indicate sampling time.

| Sample | DNA | | RNA | |
|--------|-----|---------|-------|---------|
|        | T-RFs | Shannon | T-RFs | Shannon |
| A1  | 11 | 1.55 | 23 | 2.55 |
| B1  | 11 | 1.55 | 23 | 2.23 |
| C1  | 11 | 1.49 | 22 | 2.17 |
| D1  | 11 | 1.27 | 14 | 1.70 |
| E1  | 11 | 1.48 | 20 | 2.12 |
| F1  | 12 | 1.66 | 25 | 2.55 |
| A7  | 12 | 1.74 | 28 | 2.92 |
| B7  | 12 | 1.80 | 32 | 3.09 |
| C7  | 13 | 1.77 | 27 | 2.86 |
| D7  | 12 | 1.79 | 18 | 2.46 |
| E7  | 14 | 1.89 | 30 | 2.98 |
| F7  | 13 | 1.70 | 24 | 2.70 |
| A14 | 13 | 1.70 | 36 | 3.17 |
| B14 | 12 | 1.79 | 38 | 3.09 |
| C14 | 12 | 1.83 | 33 | 2.96 |
| D14 | 12 | 1.87 | 34 | 3.08 |
| E14 | 12 | 1.83 | 32 | 2.89 |
| F14 | 12 | 1.79 | 20 | 2.18 |













**Fig. 1.** Change of methane oxidation potential in the microcosms with different
treatments. Error bar indicates standard deviation ($n$=3). Asterisk indicates the
significance between experiment group and control group ($P$< 0.05). 'ns' indicates no
significant difference among treatments at a given time.

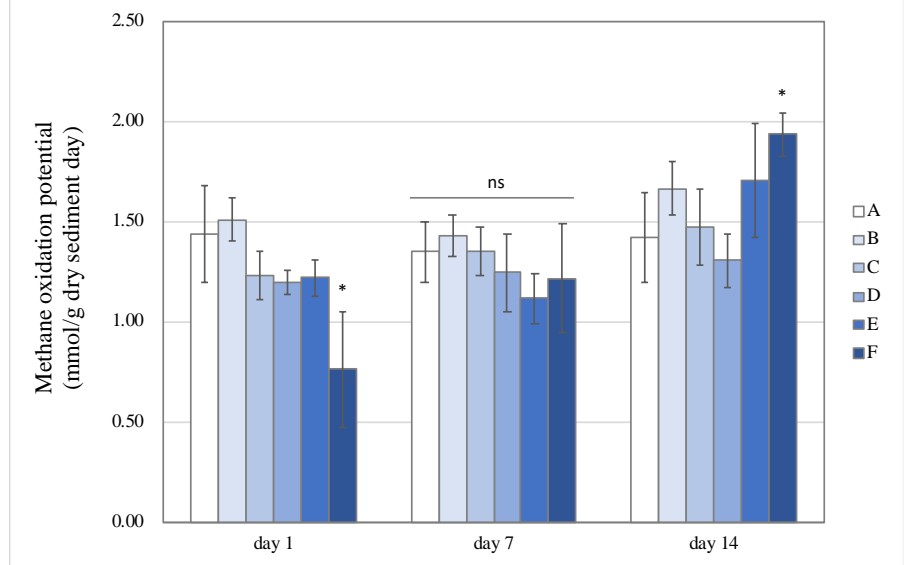




**Fig. 2.** Changes of *pmoA* gene (a) and transcript (b) abundance in the microcosms
with different treatments. Error bar indicates standard deviation (*n*=3). Asterisk
indicates the significance between experiment group and control group (*P*<0.05). 'ns'
indicates no significant difference among treatments at a given time.
**(a)**

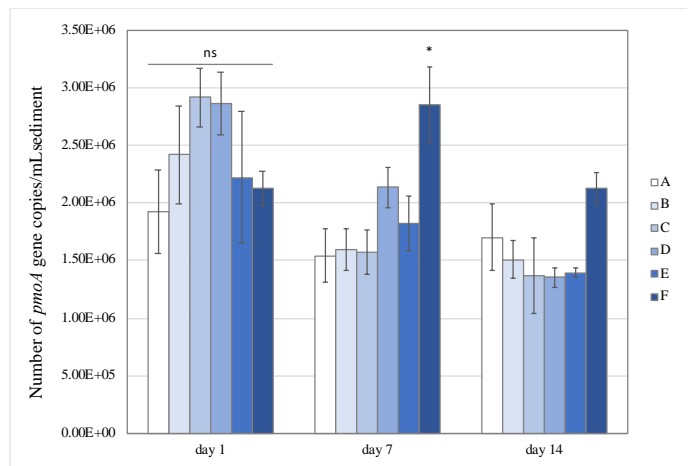



**(b)**

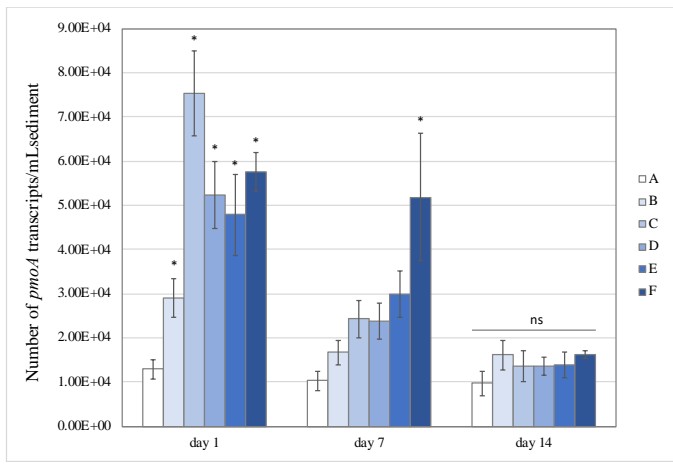







**Fig. 3.** Phylogenetic tree of obtained *pmoA* sequences and reference sequences from
the GenBank database. The predicted cut sites were shown after the accession
numbers of sequences. The dots at branches represent the support values from
bootstrap test. Branch support values of no less than 50 were dotted. The bar
represents 1% sequence divergence based on neighbor-joining algorithm.

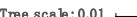

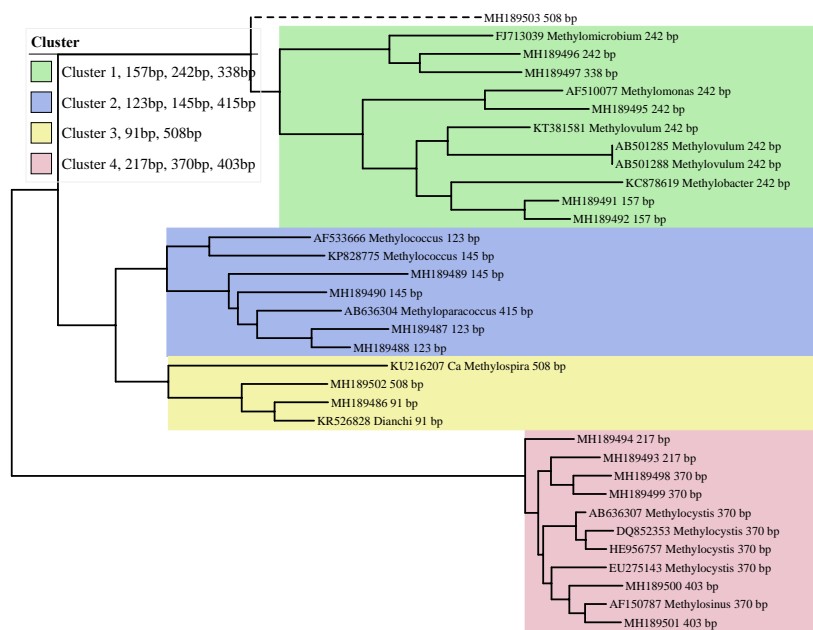








**Fig. 4.** T-RFLP profiles based on *pmoA* gene (a) and transcripts (b). For sample name,

upper case letters refer to treatment while digits indicate sampling time.

**(a)**

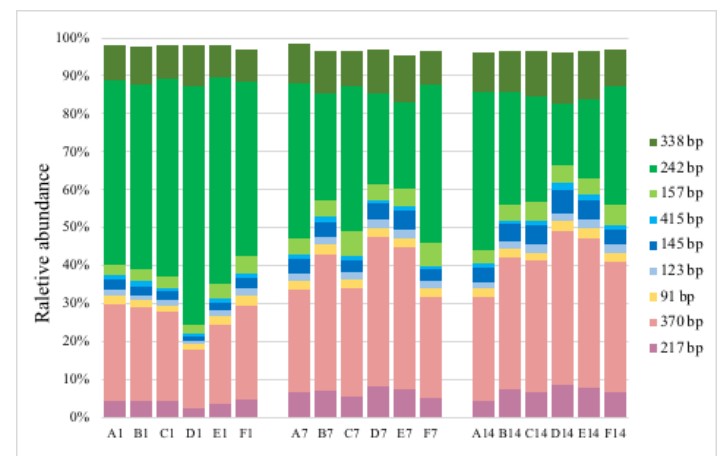


**(b)**

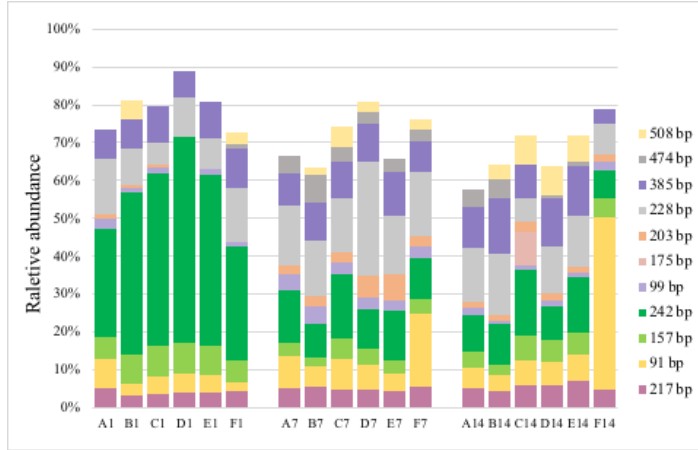











**Fig. 5.** *pmoA* gene (a) and transcripts (b)-based cluster diagrams of similarity values
for samples with different treatments. Dissimilarity levels are indicated above the
diagram. For sample name, upper case letters refer to treatment while digits indicate
sampling time.

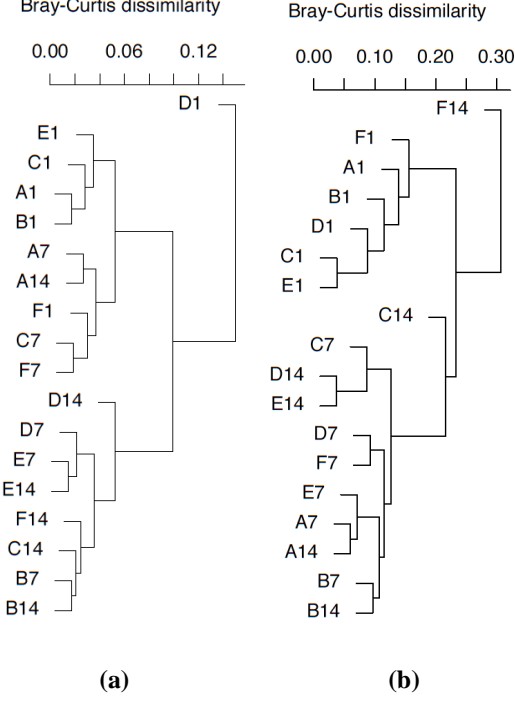


**(a)**           **(b)**




