# Peer review of "Ammonia impacts methane oxidation and methanotrophic community in freshwater sediment"

_Biogeosciences, 2018_

## Referee Comment (RC1) · Anonymous Referee #1 · 1 Jun 2018

The manuscript by Yang et al. describes the effect of different amendments of ammonium salt in sediments of a Lacustrine lake for two weeks and monitoring of gene and transcript abundance of pmoA and measurement of methane oxidation potential. Some specific comments regarding the manuscript are listed below: 1. The statement made by the authors that the impact of ammonia on aerobic methanotrophs is unclear is not true. In fact a recent study performed by Liesack group have shown that ammonia specifically inhibits high affinity methane oxidation using Methylocystis sp. strain SC2 as the model system (Dam et. al 2014). They should refer to the study and make a discussion on the high affinity methanotrophs is such environments. The authors have totally neglected the high affinity methanotrophs in their discussions.

2. The use of BciT130 for T-RFLP is also unusual. The authors must clarify the use of such an unusual restriction enzyme for generation of T-RF cuts, instead of Msp I, which have been widely used for methanotrophs.

3. In the introduction and discussion sections the authors need to mention the importance of ammonium and methane oxidation in Lacustrine environments. What physiochemical or biogeochemical evidences are there that prove the studied lake is Lacustrine in nature. In fact the term "Lacustrine" have only be used twice in the abstract and nowhere else.

4. What impact does the authors think will this study have on such lake ecosystems? Do they want to mimic some future possibilities or so? A proper objective must be developed at the beginning and the experimental design should be in sync to the objective. The different physiochemical data of the sediments must be mentioned in the result and those should be discussed in relation to the methanotroph community.

5. Use of terminologies like treatment A, B, C etc throughout the text is making the manuscript difficult to follow at times.

6. How many clone library sequences were performed? This needs to be mentioned.

7. Fig. 4: Is it a 100% graph?

---

## Author Comment (AC1) · 15 Jun 2018

The manuscript by Yang et al. describes the effect of different amendments of ammonium salt in sediments of a Lacustrine lake for two weeks and monitoring of gene and transcript abundance of pmoA and measurement of methane oxidation potential. Some specific comments regarding the manuscript are listed below: 1. The statement made by the authors that the impact of ammonia on aerobic methanotrophs is unclear is not true. In fact a recent study performed by Liesack group have shown that ammonia specifically inhibits high affinity methane oxidation using Methylocystis sp. strain SC2 as the model system (Dam et. al 2014). They should refer to the study and make

a discussion on the high affinity methanotrophs is such environments. The authors have totally neglected the high affinity methanotrophs in their discussions. Response: The authors appreciate the reviewer's suggestions. Essential information about high affinity methanotrophs has been added to the manuscript. In eutrophic lake sediment, methanogenesis was quite active, and methane was abundant. Therefore, low-affinity methanotrophs might be of greater importance in the studied lake ecosystem.

2. The use of BciT130 for T-RFLP is also unusual. The authors must clarify the use of such an unusual restriction enzyme for generation of T-RF cuts, instead of Msp I, which have been widely used for methanotrophs. Response: The authors appreciate the reviewer's suggestions. Essential discussion has been added to the manuscript. In brief, we tested all the enzymes on http://www.restrictionmapper.org with some previously retrieved pmoA NGS sequences from the same lake. In silico analysis indicates that BciT130 could generate more T-RFs, and also present a better phylogenetic resolution than MspI.

3. In the introduction and discussion sections the authors need to mention the importance of ammonium and methane oxidation in Lacustrine environments. What physiochemical or biogeochemical evidences are there that prove the studied lake is Lacustrine in nature. In fact the term "Lacustrine" have only be used twice in the abstract and nowhere else. Response: The authors appreciate the reviewer's suggestions. Revisions have been made as suggested.

4. What impact does the authors think will this study have on such lake ecosystems? Do they want to mimic some future possibilities or so? A proper objective must be developed at the beginning and the experimental design should be in sync to the ob- jective. The different physiochemical data of the sediments must be mentioned in the result and those should be discussed in relation to the methanotroph community. Response: The authors appreciate the reviewer's suggestions. Revisions have been made in the manuscript. Eutrophication is one of the major concerns in most of freshwater lakes in China. And excess ammonium input could always be observed in these

environments. Therefore, the current study is aim to investigate how methanotrophic community response to high concentration of ammonium input in lakes experiencing eutrophication process.

5. Use of terminologies like treatment A, B, C etc throughout the text is making the manuscript difficult to follow at times. Response: The authors appreciate the reviewer's suggestions. Revisions have been made to make the manuscript more readable.

6. How many clone library sequences were performed? This needs to be mentioned. Response: The authors appreciate the reviewer's suggestions. A total of 96 clones was subjected to sequencing, and 93 sequences were retrieved. This has been added to the manuscript.

7. Fig. 4: Is it a 100% graph? Response: No. Some less abundant T-RFs were not shown in the graph.

---

## Referee Comment (RC2) · Anonymous Referee #2 · 8 Jul 2018

Manuscript "No.: bg-2018-193" describes the effect of ammonia on methane oxidation and methanotrophic community in freshwater sediment. The study is interesting and the topic itself is important, but there is a major drawback. One of the main purposes of this manuscript is to investigate the effect of ammonium on community structure of aerobic methanotrophs. However, due to the methods they use, it is difficult to obtain the classification information for methanotrophs. BciT130 I striction endonuclease was used to digest purified PCR products in this study. However, the digestive enzyme often used for pmoa T-RFLP analysis is Msp I. In the present study, it was difficult to determine whether the dominant TF peak 242 bp is Methylomicrobium or Methylomonas. The main T-RF peaks in Figure 4(b), such as 385 bp and 228 bp, had no relevant information on the methanotrophic taxonomy. Compared with T-TF and clone sequencing, Miseq sequencing of pmoA gene would be a better choice for this study.

Minor comments:

Line 104 In the Materials and Methods section, methods for measuring the physical properties of sediments should be described.

Line 182: Submit the Clone sequences to NCBI and list the accession number.

Lines 597-623, in Fig 1 and 2, the methane oxidation potential increased (treatment F) or remained relatively stable (treatments C, D, E) during the incubation, however, the pmoA transcription was reduced after 14days incubation. So how could you explain the increased or stable methane oxidation by reducing pmoA transcription?

---

## Author Comment (AC2) · 20 Jul 2018

Manuscript "No.: bg-2018-193" describes the effect of ammonia on methane oxidation and methanotrophic community in freshwater sediment. The study is interesting and the topic itself is important, but there is a major drawback. One of the main purposes of this manuscript is to investigate the effect of ammonium on community structure of aerobic methanotrophs. However, due to the methods they use, it is difficult to obtain the classification information for methanotrophs. BciT130 I striction endonuclease was used to digest purified PCR products in this study. However, the digestive enzyme often used for pmoa T-RFLP analysis is Msp I. In the present study, it was difficult to determine whether the dominant TF peak 242 bp is Methylomicrobium or Methylomonas. The main T-RF peaks in Figure 4(b), such as 385 bp and 228 bp, had no relevant information on the methanotrophic taxonomy. Compared with T-TF and clone sequencing, Miseq sequencing of pmoA gene would be a better choice for this study.

Response: The authors appreciate the reviewer's valuable suggestions. We have added some discussions to explain the choice of enzyme (part 4.4). In spite of the somewhat unsatisfactory taxonomic resolution, TRFLP is still a very fast and economical approach offering an overview of methanotrophic community composition and diversity. We agree that NGS might be a better choice to get a more comprehensive profile of methanotrophic community, and will consider it in our future methanotrophic community studies.

Minor comments: Line 104 In the Materials and Methods section, methods for measuring the physical properties of sediments should be described.

Response: The authors appreciate the reviewer's suggestion. The methods used have been added in the revised manuscript (in Lines 112-116, in red).

Line 182: Submit the Clone sequences to NCBI and list the accession number.

Response: The authors appreciate the reviewer's suggestions. The accession numbers have been already listed in Figure 3 (together with the predicted T-RF length).

Lines 597-623, in Fig 1 and 2, the methane oxidation potential increased (treatment F) or remained relatively stable (treatments C, D, E) during the incubation, however, the pmoA transcription was reduced after 14 days incubation. So how could you explain the increased or stable methane oxidation by reducing pmoA transcription?

Response: The authors appreciate the reviewer's comments. However, transcripts were not directly linked to the activity. It is likely that the higher concentration of pMMO protein (resulted from the higher transcription in earlier time) maintained the relatively stable methane oxidation potential.

---

## Author Comment (AC3) · 20 Jul 2018

**Ammonia impacts methane oxidation and methanotrophic community in**

**freshwater sediment**

Yuyin Yang[1], Jianfei Chen[1], Shuguang Xie[1,*], Yong Liu[2]

[revised manuscript text omitted]

Especially, many freshwater lakes in China have been suffering from eutrophication.

The methanotrophic communities in these ecosystems have been under high ammonium pressure. The response pattern of methanotrophic community to ammonium pressure in eutrophic lakes might be different from that in oligotrophic lakes. Hence, in the present study, microcosms with eutrophic freshwater lake sediment were constructed to investigate the ammonium influence on methane oxidation potential and the abundance, transcription and community structure of aerobic methanotrophs. The object of this current study was to demonstrate how different concentrations of ammonium nitrogen impacted the structure and function of aerobic methanotrophic communities in freshwater lakes with long-term eutrophication.

**2.Materials and methods**

*2.1. Sediment characteristics*

Dianchi Lake is a large shallow lake (total surface area: 309 km$^2$; average water depth: 4.4 m) located in southeast China (Yang et al., 2016). This freshwater lake is suffering from anthropogenically-accelerated eutrophication (Huang et al., 2017).

Surface sediment (0–5 cm) (24.9286N, 102.6582E) were collected using a core sampler from the north part of Dianchi Lake in October, 2017. The pH and water dissolved oxygen (DO) were immediately measured with electrode sensors.

Ammonium nitrogen ($NH_4^+$-N) was measured using Nessler's reagent spectrophotometric method. Physicochemical properties of sediment were determined according to the literature (Wang, 2012). 
[revised manuscript text omitted]

*4.4 TRFLP fingerprinting*

T-RFLP has been a popular approach to capture microbial community diversity. It has been also widely used in community studies of methylotrophs (Mohanty et al., 2006;

Pester et al., 2004; Shrestha et al., 2010). However, the most widely used digestive enzyme, *MspI*, may not be suitable for all kinds of samples. For example, samples from littoral Lake Constance only resulted in a few T-RFs. And most of the clones from Type I were at the same T-RF length of 248 bp (Pester et al., 2004). *MspI*

enzyme also generated only very few T-RFs for the sediment samples from Dianchi

Lake. The number of T-RFs based on a certain digestive enzyme might partly depend on the in-situ microbial community features. Since T-RF was only related to the first cleavage site, if a specific point mutation got widespread (especially when the population was limited), it might notably impact the T-RF pattern. Efforts were made to avoid some of the weakness of T-RFLP and improve the taxonomic resolution, including the application of multiplex T-RFLP (Elliott et al., 2012) and the usage of primers labelled with different fluorochromes (Deutzmann et al., 2011). Here, we tried to choose a suitable enzyme based on in-silico analysis. A previous NGS result of methylotrophic community in the same region (under accession number

SRP131884) was used as a reference. The *pmoA* sequences were grouped into OTUs at 0.03 cutoff, and the representative sequences of each OTU were used in mapping.

We tested all the enzymes listed in http://www.restrictionmapper.org, and calculated the proportion of each T-RF. A simplified T-RF map (including the enzymes which could cut over 80% of total sequences) was shown in Fig S3. We expected that a good digestive enzyme should: (1) generate no or few T-RFs smaller than 50 bp or larger than 500 bp; (2) generate T-RFs having at least 2 bp differences among each other; (3)

generate more T-RFs to retrieve the diversity; (4) be consistent with taxon and phylogenetic tree (i.e., the same T-RF should not be affiliated to very distantly related taxa). Based on these rules, most of the enzymes were easy to exclude. *MspI* and

*BciT130I*, the same as *HpaII* and *EcoRII*, respectively, were further tested using a neighbor-joining tree. *BciT130I* was able to generate more different T-RFs, and thus could more ideally reflect methylotrophic diversity. It also had a better taxonomic resolution than *MspI*.

**5.  Conclusions**

This was the first microcosm study on the influence of ammonium on freshwater lake sediment methanotroph community. In freshwater lake sediment microcosm, methane oxidation potential and methanotrophic community could be influenced by ammonium amendment. Ammonia concentration had a significant impact on methanotrophic abundance and diversity, but exerted no evident influence on community structure. Compared with *pmoA* gene, transcripts were more sensitive to external ammonium addition. Further works are necessary in order to elucidate the influence of ammonium on methane oxidation in freshwater sediment.

**Conflict of interest**

The authors declare that they have no competing interests.

**Acknowledgments**

This work was financially supported by National Natural Science Foundation of

China (No. 41571444), and National Basic Research Program of China (No.

2015CB458900).

[revised manuscript text omitted]